# Enhancement of Energy Storage Performance of PMMA/PVDF Composites by Changing the Crystalline Phase through Heat Treatment

**DOI:** 10.3390/polym15112486

**Published:** 2023-05-28

**Authors:** Changhai Zhang, Xu Tong, Zeyang Liu, Yue Zhang, Tiandong Zhang, Chao Tang, Xianli Liu, Qingguo Chi

**Affiliations:** 1Key Laboratory of Engineering Dielectrics and Its Application, Ministry of Education, Harbin University of Science and Technology, Harbin 150080, China; 2School of Electrical and Electronic Engineering, Harbin University of Science and Technology, Harbin 150080, China; 3Key Laboratory of Advanced Manufacturing and Intelligent Technology, Ministry of Education, Harbin University of Science and Technology, Harbin 150080, China

**Keywords:** composite dielectric, heat treatment, poly(vinylidene fluoride), energy storage, high breakdown strength

## Abstract

In today’s contemporary civilization, there is a growing need for clean energy focused on preserving the environment; thus, dielectric capacitors are crucial equipment in energy conversion. On the other hand, the energy storage performance of commercial BOPP (Biaxially Oriented Polypropylene) dielectric capacitors is relatively poor; hence, enhancing their performance has drawn the attention of an increasing number of researchers. This study used heat treatment to boost the performance of the composite made from PMAA and PVDF, combined in various ratios with good compatibility. The impacts of varying percentages of PMMA-doped PMMA/PVDF mixes and heat treatment at varying temperatures were systematically explored for their influence on the attributes of the blends. After some time, the blended composite’s breakdown strength improves from 389 kV/mm to 729.42 kV/mm at a processing temperature of 120 °C. Consequently, the energy storage density is 21.12 J/cm^3^, and the discharge efficiency is 64.8%. The performance has been significantly enhanced compared to PVDF in its purest state. This work offers a helpful technique for designing polymers that perform well as energy storage materials.

## 1. Introduction

As technology and modern industry advance, more and more novel energy sources are being used in societal contexts, increasing the demand for energy conversion and storage technologies [1,2,3,4,5]. Compared to super-capacitors and chemical batteries, dielectric capacitors have unique advantages, such as fast charging and discharging and good power stability, and there is an urgent need for them in scenarios such as hybrid vehicles, underground resource measurement, and wind turbines. Traditional inorganic ceramic dielectric energy storage materials have a high dielectric constant, but their application is limited by their complex fabrication process and low breakdown strength. Polymer dielectric materials provide the advantages of having a high breakdown strength, being easy to make, being flexible, and exhibiting little dielectric loss; nevertheless, the limited amount of energy that can be stored in these materials is due to the materials’ poor dielectric constant [6,7,8]. For instance, the industrial dielectric material biaxially oriented polypropylene (BOPP), commonly used today, has a low dielectric constant of 2.2 and a limited energy storage density of less than 4 J/cm^3^. BOPP is frequently used because of its cheap cost and widespread availability. In this regard, the volume and weight of dielectric film capacitors need to be increased in industrial production to improve their energy density, increasing the costs. As a result, there is a pressing need to create dielectric materials with excellent energy storage capabilities and significant breakdown strengths to satisfy the scenario’s predicted future need for lightweight and tiny devices [9,10,11].

A standard dielectric capacitor’s energy density, often known as its energy storage capacity, is defined as follows:(1)$$U_{e}=\int_{P_{r}}^{P_{\max}}\mathrm{EdD}$$
where E(kV/mm), ${D(C/m}^{2})$, $P_{\max}{(C/m}^{2})$, and $P_{r}{(C/m}^{2})$ each stand for the applied electric field, the potential shift, the maximum polarization, and the residual polarization, respectively.

The symbols for the applied electric field, the potential shift, the maximum polarization, and the residual polarization, respectively, are *E, D*, $P_{\max}$, and $P_{r}$ [12]. *D*’s definition is provided by Equation (2), as:(2)$${D=\varepsilon }_{r}\varepsilon_{0}E$$

The vacuum dielectric constant is denoted by $\varepsilon_{0}$. The relative permittivity is denoted by the symbol $\varepsilon_{r}$. From Equations (1) and (2), the energy storage density of the linear dielectric can be calculated by Equation (3), as: (3)$$U_{e}=0.5\mathrm{DE}=0.5\varepsilon_{0}\varepsilon_{r}E^{2}$$

For non-linear dielectrics, the dielectric constant varies with the electric field, so the same formula as in Equation (3) cannot be obtained. Equations (1) and (2) demonstrate that the dielectric constant and the breakdown electric field govern the energy storage energy density of linear or non-linear dielectric materials. This is true whether the dielectric material is linear or non-linear [13]. Due to this, increasing the dielectric constant of dielectric materials and the breakdown electric field of these materials has become the essential way of increasing the energy storage density.

In comparison to other polymeric dielectric materials, polyvinylidene fluoride, often known as PVDF, has a relatively high dielectric constant, and as a result, it is of significant interest to researchers. In addition to possessing a high dielectric constant, PVDF and its copolymers offer remarkable chemical resilience, reasonably good crystallization, and polarization. All of these properties are important for electrical applications [14]. On the other hand, the relatively high polarization strength and dielectric loss of PVDF severely restrict the escape of stored energy during charging and discharging. This results in low charging and discharging efficiency and significant heat losses, severely limiting their application space and progression prospects. Doping inorganic nanoparticles into PVDF is the most popular technique for improving the performance of the polymer [6,15,16,17,18,19,20]. Polymer-based mixed media with high energy storage properties can be prepared by considerably increasing its dielectric constant. For example, Jian et al. prepared BT@TO two-dimensional nanosheets as fillers to prepare PVDF-based composites with enhanced dielectric constants and obtained an ultra-high energy density of 21.3 J/cm^3^ and a 61% discharge efficiency under an electric field of 550 kV/mm [21]. Incorporating inorganic components with a high dielectric value into polymer composites allows for a straightforward and speedy method of raising the dielectric properties of the resulting material. However, the charge accumulation and electric field distortion that can occur as a result of a mismatch in the dielectric constant between inorganic and organic phases, the agglomeration phenomenon that can occur as a result of the uneven dispersion of inorganic fillers, and the structural defects at the interface can all reduce in the composites’ breakdown durability and energy storage capacity [3,10,15,22]. At the same time, the interface polarization caused by doping inorganic materials will also produce significant losses, reducing the material charge and discharge efficiency. Blending multiple polymers to combine their properties is also a common and effective means to enhance the performance of dielectric materials [23,24,25]. Since PVDF is a non-linear dielectric material with a significant dielectric loss, researchers often add linear polymers with a low dielectric loss and a high breakdown strength. This is performed to increase the polymer’s discharge efficiency and its breakdown strength. For illustration, Chi et al. produced composite dielectric materials made of PMMA and PVDF by combining the two materials. The PMMA/PVDF binary blended composite containing 50% PMMA has a storage density of 20.1 J/cm^3^ and a discharge efficiency of 63.5% when subjected to an electric field of 570 kV/mm. This provides the material with a good capability for the storage of energy [26].

A range of non-polar and polar crystalline phases may occur in PVDF since it is a semicrystalline ferroelectric polymer. The symbols α, β, γ, δ, and ε refer to these phases. PVDF’s most frequent crystalline phases are α, β, and γ. The β-phase is highly polar because it is in the all-trans conformation (TTT), while the γ-phase is in the trans conformation (T3GT3G’) and has low polarity. Moreover, the structure denoted by the symbol α is known as a trans-trans-intertwisted (TGTG’) structure, which lacks polarity [13,27,28]. The prominent dipole moments of the β- and γ-phases in PVDF lead to high dielectric constants, but at the same time, the significant electric dipole moments of the β-phases also carry large residual polarization and dielectric losses for PVDF, resulting in a low charging and discharging efficiency. It has been shown that the phase structure inside PVDF can be changed by annealing and quenching, stretching, and pressing, thus changing the ratio of α, β, γ, or the crystal volume to improve the overall functionality of PVDF [13,29,30].

This study anticipates that mixing the PMMA with PVDF will boost the breakdown strength of the PVDF while concurrently decreasing the material’s dielectric loss. The phase structure of PVDF was changed by subjecting it to heat treatment, and the researchers looked at how the phase transition of PVDF and the content of PVDF/PMMA blends affected the characteristics of the blends. The final energy storage density of 21.12 J/cm^3^ and the efficiency of 64.8% were obtained at a processing temperature of 120 °C, at a ratio of 5.5 to 4.5 of PVDF blended with PMMA. According to the study, proper heat treatment of the PVDF-blended polymer can effectively enhance its performance. This provides a new path for future research and development of dielectric energy storage materials.

## 2. Results and Discussion

This study used an SEM to determine how adding varying proportions of PMMA affected the microstructure. Figure 1 shows that all the films had a thickness of about 15 μm and a uniform texture without fractures. As can be seen from Figure 1a,f, there was not a notable change in the microstructure of the single substances, and they both exhibited a homogeneous and continuous phase of texture. This can be deduced from the fact that the microstructure is visible in both cross-sections. According to Figure 1b–e, the PVDF and PMMA blended films showed no apparent holes and defects after doping with different proportions of PMMA, and the structure was uniform and continuous without phase separation and new phase generation. When the pictures of the blended films and those of the pure films were compared, there was no noticeable variation in the microscopic morphology of either kind of film. This demonstrates that PVDF and PMMA have high miscibility and compatibility. Previous research [31,32,33] found that the two polymers, PVDF and PMMA, have very similar solubility parameters of 22.7 and 23.2 [34], respectively. Moreover, their compatibility with each other is excellent. 

Figure 2 illustrates how the crystal structures of PVDF/PMMA blends with various PMMA content were examined using the XRD method. When looking at pure PVDF, the diffraction peaks at 17.9° (100), 18.31° (020), and 20.20° (021) corresponded to the nonpolar α-phase. PVDF’s weak polar γ-phase displayed diffraction peaks at 18.50° (020) and 20.1° (110). PVDF’s polar β-phase exhibited a diffraction peak at 20.9° (110) [34,35]. The combined reflection of the (201) and (111) planes of the β-phase, which was brought on by molecular flaws brought on by head–head and tail-tail (HHTT) sequences, characteristically exhibited another peak at around 40° [36]. A peak in amorphous dispersion may be seen for PMMA at 12°. When the quantity of PMMA blending grew, the relative strength of the distinctive peak associated with PVDF dropped, and the diffraction peak with a broad peak between 18° and 40° progressively vanished [37]. As shown in Figure 2, there was some enhancement in the β-phase and less pronounced changes in the α-phase and γ-phase. According to the findings of the XRD experiment, the presence of PMMA had a discernible influence on the crystalline phase of PVDF. This may result in a decrease in the crystallinity of PVDF. The explanation for this phenomenon is that the breakdown of amorphous PMMA into short sequences of crystalline areas, which results in decreased crystallinity, acts as a barrier to creating long sequences of PVDF polymer, leading to reduced crystallinity. 

For the impact of PMMA concentration on the structure of blended composites to be better understood, infrared spectra of blended composites were collected between 450 cm^−1^ and 3150 cm^−1^. Figure 3 shows that the ferroelectric polymer PVDF displayed a unique IR characteristic absorption at 611 cm^−1^ and 966 cm^−1^ attached to the α-phase. The absorption peak at 881 cm^−1^ corresponds to the symmetric stretching vibration of the -CF_2_-bond, and the absorption peak at 1073 cm^−1^ corresponds to the symmetric stretching vibration of the -CF_2_- bond, both of which are part of the β-phase. The γ-phase is connected to the weak absorption peaks that may be shown at 512 cm^−1^ and 812 cm^−1^ [26]. The peak around 840 cm^−1^ is often connected with the β- and γ-phases. The high absorption values at 989 cm^−1^ and 1449 cm^−1^, respectively, are equivalent to the methoxy (-O-CH_3_-) bending and stretching vibrations of PMMA. Moreover, the PMMA carboxyl stretching vibrations correlate to the absorption maxima at 1270 cm^−1^ and 1729 cm^−1^, respectively. The absorption peaks at 2845 cm^−1^, 2955 cm^−1^, and 3000 cm^−1^, respectively, indicate the methylene (-CH_2_-) and methyl (-CH_3_) of the PMMA [26,34]. Following the IR spectrum analysis outcomes, the IR sharp peaks of PMMA developed in the blended composites and consistently increased in intensity as the amount of PMMA used to make the composites in-creased. On the other hand, the IR characteristic peaks of PVDF gradually weakened. It can also be seen that when the percentage of the PMMA component in the material grew, the intensity of the absorption peak in PVDF equal to the α-phase decreased. Compared to PVDF that has not been mixed with PMMA, the relative concentration of PVDF’s corresponding β-phases at 745 cm^−1^ and 989 cm^−1^ has been shown to increase when the PMMA content increases.

With the use of the DSC technique, we were able to explore the crystallinity as well as the melting point of the all-organic PMMA/PVDF films. As shown in Figure 4, the melting point of pure PVDF was 169 °C, and the glass transition temperature of pure PMMA was 105 °C. The findings in Figure 5 demonstrate that the melting temperature point of PMMA/PVDF films gradually decreased as the proportion of PMMA present in the film increased. Its melting point dropped to 159 °C when 60% more PMMA was added than before.

The crystallinity can be calculated by Equation (4) [38]:(4)$$X_{c}=\frac{{\Delta H}_{m}}{{\Delta H}_{m}^{*}}\times 100\%$$

It is possible to determine the heating enthalpy of the mixed composite, represented by the symbol $\Delta H_{m}$. This may be accomplished by combining the peak shape area of the DSC data. It has been shown that the symbol represents the melting heat for PVDF with 100% crystallinity and that this value was 104.7 J/g.

The crystallinity of each component was obtained by calculation, where the crystallinity of pure PVDF was 32.55%, the crystallinity of PVDF/PMMA = 6:4 was 25.9%, the crystallinity of PVDF/PMMA = 5.5:4.5 was 15.69%, the crystallinity of PVDF/PMMA = 5:5 was 13.19%, and the crystallinity of PVDF/PMMA = 4:6 was 12.09%. The results show that the crystallization of PMMA/PVDF all-organic films decreased with the increasing PMMA concentration. This is because the macromolecular restriction of PMMA fragments on the PVDF entanglement spacing or chain environment caused the crystallization rate to decrease. Since an increase in the PMMA concentration had a diluting impact, there is a possibility that the crystallinity of the material may diminish.

The dielectric properties of mixed composite dielectric materials containing different quantities of PMMA were the subject of research. Figure 6 illustrates that when the frequency was increased, the dielectric constant of PVDF, a ferroelectric polymer with a polycrystalline phase, rapidly dropped. This can be observed when looking at the graph [13]. This is because the dipole of PVDF does not follow the frequency change at high frequencies, and the dipole shift polarization is not fully established in time at high frequencies. PMMA, a linear polymer, has a substantially lower dielectric constant than PVDF, and its frequency-dependent dielectric constant change is relatively mild. The dielectric constant of PMMA blended with PVDF is in the middle of that of PVDF and PMMA. This is because PMMA has a low dielectric constant, and the blending process alters the crystallinity of PVDF. PVDF-based composites have dielectric losses that peak at high and low frequencies due to dipole orientation polarization or ion polarization at low frequencies and amorphous phase molecular chains at high frequencies. Doping PVDF with PMMA results in a reduction in the crystallinity of the material and a reduction in the crystalline area. As a result of the reduction in the crystalline area, it is now much simpler for the dipole to switch to polarization, which decreases the complex’s dielectric loss. Additionally, there is some interaction between several functional groups on the main chain of the molecular chain of PMMA and PVDF. This interaction could restrict the mobility of the dipole, which would ultimately result in a decrease in the amount of dielectric loss [26,39].

As shown in Figure 7, the carriers acquired a more significant amount of energy when subjected to the intense electric field, and the current density tended to rise in tandem with the increasing magnitude of the applied electric field. The leakage current density of PMMA, and PVDF/PMMA composites, was much lower than that of pure PVDF by approximately two orders of magnitude when subjected to an electric field with an intensity of 100 kV/mm. The results showed that the leakage current density could be reduced, and the breakdown electric field strength of the composites could be increased by adding PMMA to the blended composites. This might be because PMMA alters the internal structure of the composite film and decreases the frequency of interchain defects, both of which work to restrict the mobility of carriers. In addition, the interfaces between the different domains of PVDF and PMMA composites might act as potential hurdles for carriers, which reduces the likelihood that carriers will be able to cross these barriers [36,40,41]. Therefore, PMMA/PVDF has poor electrical conductivity.

The performance, dependability, and operating boundaries of polymers’ energy storage systems are significantly influenced by their breakdown strength. In this paper, the breakdown performance of the PVDF/PMMA copolymer was evaluated by Weibull distribution, which can be obtained by Equation (5) [42,43]:(5)$$P(E)=1-\exp(-{{(E/E}_{b})}^{\beta })$$

With a failure probability of *P* = 63.2%, the cumulative failure probability is denoted by the symbol *P*(*E*), the experimental breakdown strength is denoted by the letter *E*, and the symbol denotes the characteristic breakdown strength. The characteristic breakdown strength is denoted by the symbol $E_{b}$. The *β* shape parameter is used to assess the dispersion level in the data. 

If the inverse is taken for both sides of the above Equation (5), the equation becomes a linear regression equation, as follows in Equation (6):(6)$$\mathrm{In}[-In\left(1-P\left(E\right)\right)]=\beta (\mathrm{InE}-{\mathrm{InE}}_{b})$$

The value of *E_b_* can be calculated by Equation (5), and the corresponding value of *P*(*E*) when assigned an electric field can be obtained from the following Equation (7):(7)$$P\left(E\right)=\frac{i-0.5}{n+0.25}$$

In the above equation, *i* is the order of the values of breakdown strength, *E*, from smallest to largest, and *n* is the number of samples taken for each test.

According to the data in Figure 8, the composite material exhibited a progressive improvement in its breaking strength as the proportion of PMMA in the material increased. When the proportion of PVDF to PMMA in the material was 5.5:4.5, the breakdown electric field strength achieved its highest value of 709.5 kV/mm. This number is much greater than the breakdown strength of PVDF. However, it is below the breakdown electric field strength of PMMA. First off, the breakdown electric field strength of the PMMA and PVDF mixed composite polymer may be significantly increased due to the linear polymer’s superior breakdown electric field strength. The stacking and tangling of molecular chains may also significantly affect the breakdown strength. The poor stacking structure in the amorphous portion of the polymer, which results in a large free volume and disordered structure in the amorphous section, is demonstrated to cause micropores between the polymer chains and an increase in the average free journey. As a result, the carriers build up more energy in the field, making overcoming the probability barrier easier. The composite’s tensile strength is then increased to a greater degree [36,40,44]. PMMA has a strong Young’s modulus, which might boost the mechanical characteristics of the mixed composites and raise their resistance to electrical breakdown. PMMA also has a high electrical breakdown resistance. Nevertheless, inserting excessive PMMA increases flaws within the blended composites, decreasing the blended composites’ breakdown strength [26].

For researchers to investigate the potential for energy storage of PMMA/PVDF mixed polymers, the hysteresis lines of PMMA and PVDF mixed polymers with variable quantities of PMMA contained in them were tested. The quantity of PMMA that was included in the blend polymers varied. When PVDF was exposed to the electric field intensity of 388 kV/mm, the material produced a sizeable maximum polarization ($D_{\max}$~7.52 μC/cm^2^), as shown in Figure 9. On the other hand, the residual polarization was also quite large ($D_{r}$~2.10 μC/cm^2^), which would result in a significant loss of energy and heat and a reduction in charge/discharge efficiency. This is because PVDF is a fluorinated ferroelectric polymer with the formula -(CH_2_-CF_2_)_n_- [8,13].

The highly polar C-F bond, the primary building block for the dipole to switch to polarization, is produced by the fluorine and carbon atoms’ differing electronegativities. PVDF’s primary crystalline phases are the α-, γ-, and β-phases. The γ-phase of PVDF exists as weakly polarized, while α-phase is not macroscopically polarized. Moreover, polarity in the β-phase is vital. A relaxation process occurs, and a significant residual polarization is produced when the polar phase is rotated while being affected by the electric field [45]. PMMA, as a linear polymer, exhibits a very narrow hysteresis line compared to PVDF, which has significant maximum and residual polarization. At an electric field strength of 809 kV/mm, maximum polarization in PMMA was 4.52 μC/cm^2^, and residual polarization in PMMA was 0.83 μC/cm^2^. The electric hysteresis line of the co-blended film dramatically shrank when PMMA was added to PVDF, and an increase in PMMA concentration accompanied this shrinking. At the PVDF:PMMA ratio of 5.5:4.5, a maximum polarization ($D_{\max}$) of 7.1 μC/cm^2^ and a residual polarization ($D_{r}$) of 1.38 μC/cm^2^ were achieved at an electric field strength of 809 kV/mm. Between pure PVDF and pure PMMA, both the maximum and residual polarization decreased. Compared to pure PVDF, the dielectric loss decreased, and the charging and discharging efficiency increased.

Figure 10 shows the results of testing the PVDF/PMMA blend’s ability to store energy. The discharge energy density was obtained from the discharge curve area in Figure 9, according to Equation (1), and the energy storage efficiency was obtained from the ratio of the charge curve area to the discharge curve area. At an electric field of 389 kV/mm, PVDF obtained an energy storage density of 7.47 J/cm^3^ and a discharge efficiency of 42.10%. This behavior was influenced by an inadequate breakdown strength, large dielectric loss, and a significant residual polarization of PVDF [46]. The linear polymer PMMA had a muscular breakdown strength, an energy storage density of 14.6 J/cm^3^, and an energy storage efficiency of 73.47% at an electric field strength of 809.651 kV/mm. Nevertheless, the low dielectric constant and polarization prevented the energy storage density from increasing, and under the same circumstances, PVDF had a far higher energy storage density than PMMA. Even though the electric field intensity between the two materials was the same, this was still the case. When PVDF was combined with PMMA, the performance of the blended polymer was improved. The combined polymer had a higher energy storage density than PMMA and a substantially better breakdown strength and discharge efficiency than PVDF. The PVDF:PMMA ratio of 5.5:4.5, with an electric field strength of 689.7 kV/mm, had an energy storage density of 17.6 J/cm^3^ and a 63% efficiency.

According to the study’s findings, the crystalline phase structure and composition of PVDF may be altered by using heat treatment to improve the material’s capacity for energy storage. The PVDF:PMMA = 5.5:4.5 blended films were put through thermal processing at temperatures of 90 °C, 120 °C, and 150 °C, respectively, in light of the study provided earlier. The scanning electron microscope investigated how the blended films’ microstructure altered when subjected to varying temperatures. Figure 11 demonstrates that the thickness of each sample was about 15 μm, and the cross-sectional morphology was flat and homogeneous, without any visible phase separation. The findings indicate that the cross-section of each sample is virtually entirely composed of a single-phase system. The scanning electron micrographs of the samples subjected to heat treatment at several temperatures revealed no discernible differences, suggesting that no thermally induced phase separation occurred under the heat treatment settings in this investigation.

The XRD method was used to investigate how the thermal treatment influenced the crystal structure of the co-blended films. Figure 12 demonstrates that diffraction fronts for some different phases may be detected at the three treatment temperatures of 90 °C, 120 °C, and 150 °C. The α-phase of PVDF contains three peaks of diffraction that are positioned at corresponding angles of 17.9° (100), 18.31° (020), and 20.20° (021). The diffraction peaks that occur at angles of 18.50° (020) and 20.10° (110), respectively, correspond to the weakly polarized γ-phase of PVDF. The diffraction peak at 20.9° (110) is associated with the polar β-phase of PVDF [13,27].

The diffraction peak of the β-phase and the γ-phase was steadily improved, and that of the α-phase was gradually decreasing when the temperature of the heat treatment was increased from 90 ℃ to 150 ℃. According to the findings, the heat treatment altered both the crystal structure as well as the crystalline phase composition of the co-blended film. As the temperature of the heat treatment was increased, the nucleation of γ-phase PVDF occurred more quickly; as a result, the γ-phase that is dominant in the blended film can endure higher temperatures for a tremendous amount of time [36,37,40].

To better understand the impact of the heat treatment on the structure of the blended composites, infrared spectra were collected in the spectrum of 450 cm^−1^ to 3150 cm^−1^. Figure 13 shows that the blended polymers displayed typical absorption peaks of the α-phase IR at 611 cm^−1^ and 966 cm^−1^. These peaks may be seen in the blended polymers. Between 881 cm^−1^ and 1073 cm^−1^, the β-phase absorption reached its maximum value. At 840 cm^−1^, an asymmetric stretching vibrational absorption peak of -CF_2_- was seen, as commonly seen in the β- and γ-phases. The γ-phase connects the weak absorption peaks at 512 cm^−1^ and 812 cm^−1^. It is possible to see the PMMA absorption peak, as mentioned in the preceding section [34,40,44]. The IR spectrum findings revealed that when more heat treatment was introduced, the β-phase and γ-phase IR characteristic peaks of PVDF in the blended composites progressively grew, while the α-phase IR characteristic peaks of PVDF gradually diminished. This demonstrates how the heat treatment altered the blended films’ crystal structure and crystalline phase composition. With an increase in the heat treatment temperature, the α-phase progressively transitioned into the γ-phase and the β-phase, changing the relative proportions of each phase. The IR results can correspond to the XRD results.

Figure 14 shows the DSC results of the mixes after the heat treatment. The graph demonstrates how the melting point of the blended polymers gradually increased from 161 °C to 165 °C as the temperature of the heat treatment increased. The crystallinity of each component was calculated by Equation (3), where the crystallinity of the 90 °C treatment temperature was 13.42%, the crystallinity of the 120 °C treatment temperature was 15.70%, and the crystallinity of the 150 °C treatment temperature was 20.68%. The calculations showed that higher treatment temperatures would result in more excellent crystallinity.

Figure 15 shows the dielectric characteristics of the co-blended films consisting of PMMA and PVDF in the ratio of 5.5:4.5 after being subjected to varying temperatures throughout the heat treatment process. The dielectric constant of blended composites may diminish, as seen in the image, as the frequency increases. This is because a high-frequency alternating electric field does not cause the dipole orientation’s polarization to shift [47]. Additionally, it has been shown that the co-blended composite’s dielectric constant increased as the heat treatment temperature increased. Due to the composite film’s greatest crystallinity during the 150 °C heat treatment, this treatment yielded the blended film’s highest dielectric constant. The dielectric loss dropped and rose with increasing frequency, as shown in the figure. 

First, for pure PMMA, two relaxation peaks of very low intensity were exhibited at around 100 Hz and 5 MHz, respectively, as highly rigid chain segment motion (α-relaxation) and dipole rotational motion (β-relaxation). β-relaxation peaks were obtained due to the rotation of the ester functional group (-COOCH_3_) around the C-C bond, connecting it to the repeating unit of the PMMA backbone [36]. For PVDF, two (half) relaxation fronts were observed at low and high frequencies, respectively. The relaxation front near 10 Hz at low frequencies (α_c_ relaxation) is a dipole relaxation along the α-type chain axis, and amorphous dipoles and dipoles cause the relaxation front around 1 MHz (α_a_ relaxation) at the crystal/amorphous interface. The relaxation of PVDF was significantly suppressed after PMMA with PVDF, with the increase of the PMMA blending concentration. In contrast, the crystallinity of PVDF gradually increased with the increase in the heat treatment temperature. The increase of crystallinity made the crystalline α_c_-relaxation and the dipole relaxation at the crystal/amorphous interface increase simultaneously, so the highest degree of relaxation was observed at the treatment temperature of 150 °C [48].

The low-leakage current density of the composites contributed to an improvement in both the breakdown field and the energy storage density of the materials. Figure 16 displays the blended materials’ leakage currents after the heat treatment. The current densities of the films after heat treatment at 90 ℃ and 120 ℃ were much lower than those after treatment at 150 ℃, while the current density after heat treatment at 120 ℃ was slightly lower than that after heat treatment at 90 ℃. When the thermal motion breaks the potential barriers between the segments and domains of PMMA and PVDF of molecules brought on by high temperatures [36,40,41], the carriers may travel more freely, improving the current density.

Figure 17 shows the breakdown field intensity of the PVDF:PMMA blended film according to the Weibull distribution features at different heat treatment temperatures. The most considerable breakdown field strength was attained at the treatment temperature of 120 °C, and it tended to grow, then decrease, as the temperature increased. There are several reasons for this. Firstly, the high-temperature treatment reduced the voids and defects between PVDF and PMMA, and the quality of the film was effectively improved; secondly, the high-temperature heat treatment caused the existence of a specific difference in the dielectric constant between PVDF and PMMA, resulting in a local electric field that reduced the breakdown field. In addition, the high temperature of 150 ℃ promoted molecular thermal motion, which made the molecular chains of PVDF and PMMA move. The molecular structure may be disrupted, making the breakdown field strength lower.

The D-E curves of the PVDF:PMMA = 5.5:4.5 blended films are shown in Figure 18 at the heat treatment temperatures of 90 °C, 120 °C, and 150 °C. The highest maximum polarization ($D_{\max}$~8.42 μC/cm^2^) was obtained at 150 ℃, but the residual polarization was also considerable ($D_{r}$~1.92 μC cm^−2^). Only when the temperature of the heat treatment was 120 ℃ was it possible to produce a greater maximum polarization and a more negligible residual polarization ($D_{\max}$~μC/cm^2^, $D_{r}$~1.48 μC/cm^2^). As the temperature of the heat treatment was raised to a higher level, there was a corresponding modest rise in the maximum polarization value of the composites. This is because greater temperatures cause a gradual change from the α-phase to the γ-phase and the β-phase, which increases the polarization intensity [26,37,40]. The dielectric and dielectric loss diagrams are related to the rise in temperature since it also causes an increase in crystallinity, which boosts the maximum and residual polarization. At a temperature of 120 ℃, it was possible to produce good maximal polarization and excellent residual polarization, and this indicated that a good energy storage effect would be attained.

Figure 19 depicts the performance of PMMA:PVDF = 5.5:4.5 blended films in energy storage when subjected to varying temperatures throughout the heat treatment process. The highest performance was found at a treatment temperature of 120 °C, with an energy storage density of 21.12 J/cm^3^ and a discharge efficiency of 64.8%. The other two treatment temperatures were 90 °C and 150 °C. The energy storage density increased by 300% compared to pure PVDF, while the efficiency increased by 50%. In addition, the breakdown field strength increased to 729.42 kV/mm from a previous value of 389 kV/mm. In the context of this study, a composite film that was heat-treated at 120 °C exhibited an excellent energy storage performance, maximum polarization, and the ability to withstand a higher applied electric field than pure PVDF. Furthermore, the composite film’s performance was significantly improved compared to that of pure PVDF.

## 3. Materials and Methods

### 3.1. Materials

The 3F-Wanhao Fluorine Chemical Co., Ltd., Inner r Mongolia autonomous regions, China, supplied polyvinylidene fluoride, also known as PVDF. Fuyu Fine Chemical Co., Ltd., Tianjin, China, provided N,N-dimethylformamide (DMF, also known as C_3_H_7_NO). The company Sinopharm Chemical Reagents Co., Ltd., Shanghai, China, supplied polymethyl methacrylate, also known as PMMA.

### 3.2. Preparation

The PMMA/PVDF blended composites were made utilizing a solution blending approach as the primary preparation mode. In a beaker that had previously been filled with a certain volume of DMF solution, the right amounts of powdered PVDF and PMMA were measured and added to the mixture. After stirring for twelve hours at room temperature, the beaker was removed from the magnetic stirrer. After that, the combined solution was poured onto a glass plate and dried in a vacuum oven for ten hours at a predetermined heat treatment temperature to speed up the process of the DMF evaporation. In the last step, the glass plate was peeled away to reveal a completely organic composite film with a consistent texture.

SEM test sample preparation: The samples were prepared at 5 mm × 10 mm in size, soaked in liquid nitrogen for 30 min, then brittle-fractured, sprayed with gold for 120 s, and placed on the sample bench for testing with SEM.

Dielectric test sample preparation: The sample was prepared at 15 mm × 15 mm in size, vaporized with a 9 mm-diameter aluminum electrode, and the dielectric test was performed at room temperature.

Breakdown, leakage current, and ferroelectric test sample preparation: The sample was prepared at 6 mm × 6 mm in size, vaporized with a 3 mm-diameter aluminum electrode, and testing was performed at room temperature.

XRD and IR sample preparation: The sample was directly cut into 20 mm × 20 mm in size and placed in IR and XRD testers for testing.

### 3.3. Several Techniques of Characterization

An X-ray diffractometer (XRD) made by PANalytical in the Almelo, Netherlands was used to analyze the crystal structure of the all-organic composite films. The step size of the instrument’s scanning angle was 0.2°, and its scanning angle varied from 10° to 50°. Research on the molecular structure of organic composite films and their functional groups and chemical bonds was carried out with a Fourier transform infrared spectrometer with the model number EQUINOX55 (Bruker, Karlsruher, Germany). We used broad-band dielectric spectra supplied by Novocontrol, Montabaur Germany, to evaluate the dielectric properties of the all-organic composite films. An investigation of the cross-sectional morphology of organic composite films was carried out with the assistance of a scanning electron microscope (SEM) developed and produced in Tokyo, Japan by Hitachi. The next set of tests was carried out by evaporating aluminum electrodes on both sides of the organic composite film using ZHD-400 high vacuum evaporation coating equipment that was manufactured by Beijing Techno Technology Co., Ltd, Beijing, China. Using a Radiant Premier II ferroelectric meter, Radiant Technologies. Inc, Albuquerque, New Mexico, United States. we measured the potential shift–electric field (D-E) curves and the current–voltage (I-V) parameters of the all-organic composite films at a frequency of 10 Hz.

Differential scanning calorimetry was used so that both the melting and crystallization processes of the blended composites could be studied and understood (DSC, METTLER TOLEDO, Zurich, Switzerland). The DSC measurements were performed in an environment containing nitrogen at temperatures ranging from 20 to 200 °C at a rate of 10 °C per minute, with a sample mass of about 7 milligrams inside standard aluminum capsules.

## 4. Conclusions

In this research project, PMMA/PVDF mixed composite film production was accomplished using solution casting. The PMMA doping ratio and heat treatment temperature influencing the microstructure and energy storage capacity of PMMA and PVDF blends were methodically explored. It was possible to obtain the ideal ratio of PMMA:PVDF = 5.5:4.5 by modifying the introduction of PMMA. Heat treatment was also used to regulate the content and the structure of the crystal phase. At a temperature of 120 °C, the best performance was obtained with an energy storage density of 21.12 J/cm^3^ and a discharge efficiency of 64.8%. The development of this system offers a practical means of enhancing polymer matrix composites’ capacity for energy storage.

## Figures and Tables

**Figure 1 polymers-15-02486-f001:**
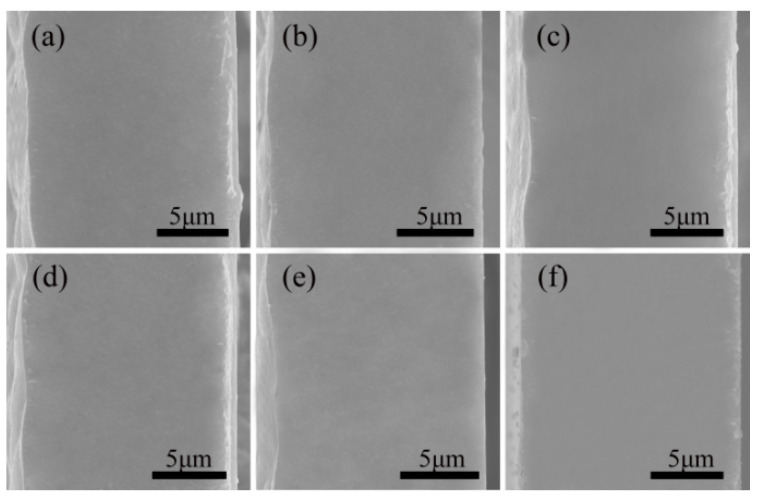
Image captured by a scanning electron microscope (SEM) of PVDF and PMMA blended polymers with different ratios: (**a**) pure PVDF, (**b**) the ratio of PVDF:PMMA was 4:6, (**c**) the ratio of PVDF:PMMA was 5:5, (**d**) the ratio of PVDF:PMMA was 5.5:4.5, (**e**) the ratio of PVDF:PMMA was 6:4, and (**f**) pure PMMA.

**Figure 2 polymers-15-02486-f002:**
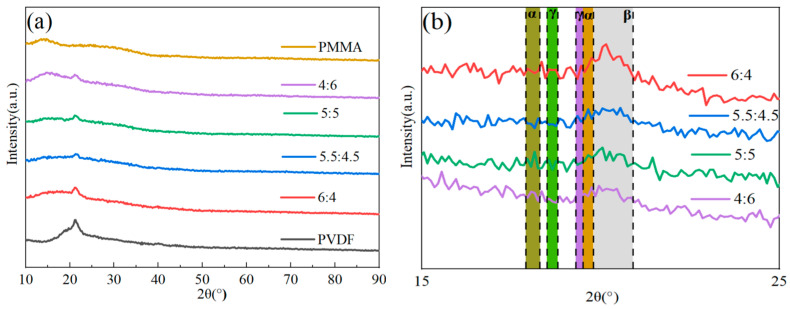
XRD patterns of PMMA and PVDF mixed composite films with different contents. (**a**) XRD images of each component. (**b**) Local enlargements of some components.

**Figure 3 polymers-15-02486-f003:**
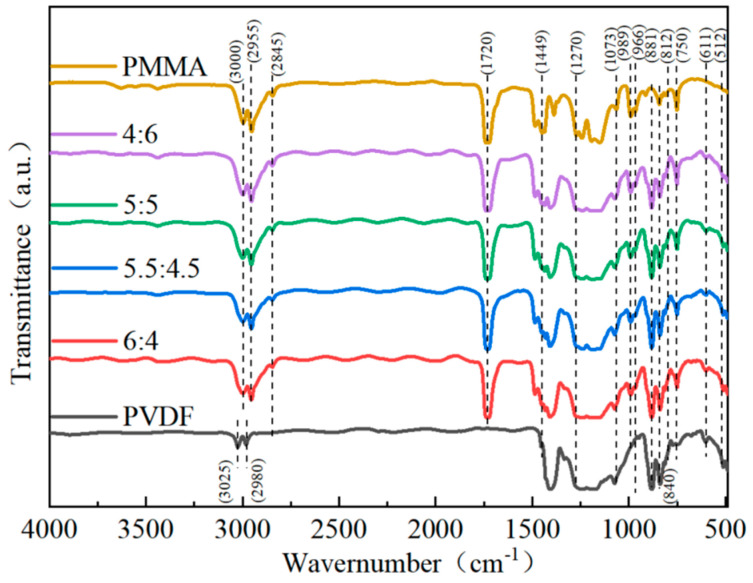
FTIR patterns of microstructures of PMMA/PVDF blended composite films with varying amounts of PMMA in their compositions.

**Figure 4 polymers-15-02486-f004:**
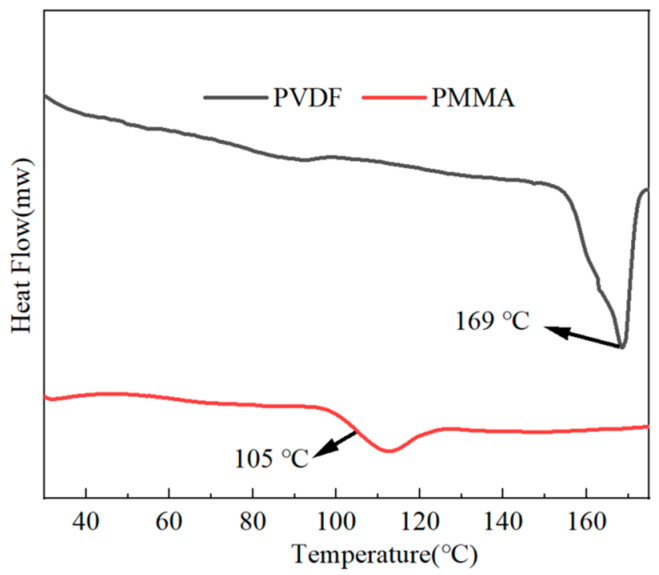
DSC curves of PVDF and PMMA.

**Figure 5 polymers-15-02486-f005:**
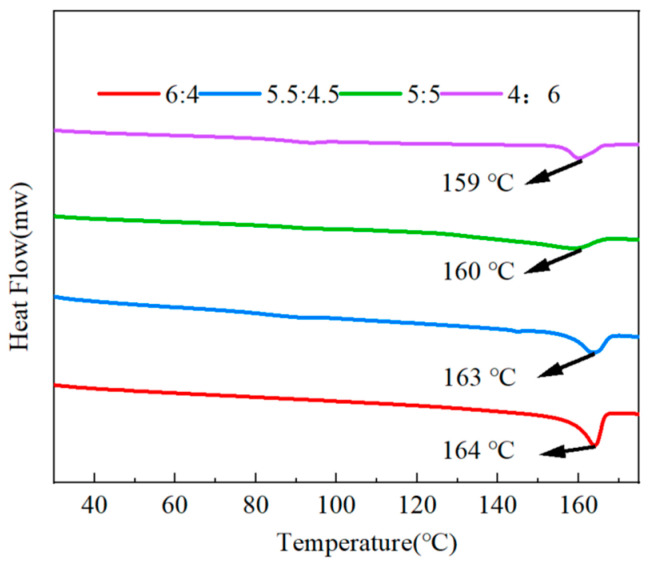
PMMA/PVDF all-organic films with different PMMA contents, as measured by DSC curves.

**Figure 6 polymers-15-02486-f006:**
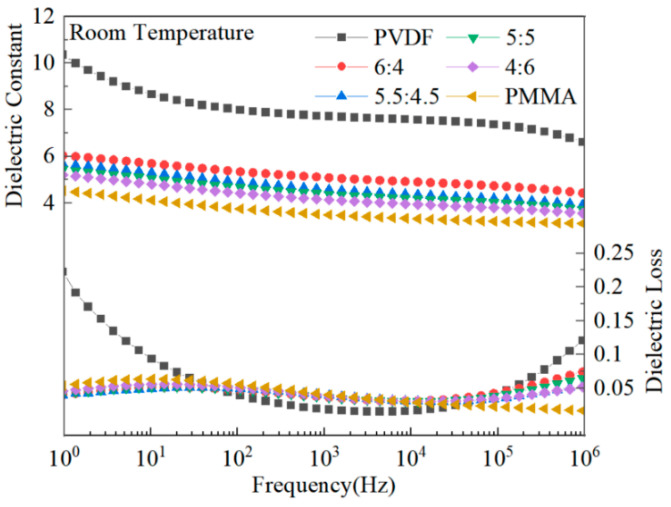
The dielectric characteristics of the various components that make up PMMA and PVDF.

**Figure 7 polymers-15-02486-f007:**
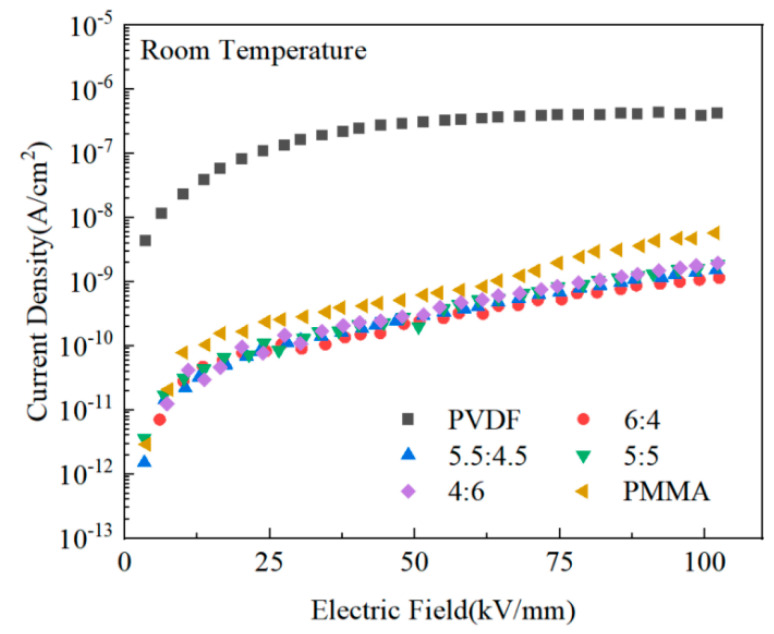
Comparison of the leakage current densities of PMMA and PVDF blends with varying amounts of PMMA.

**Figure 8 polymers-15-02486-f008:**
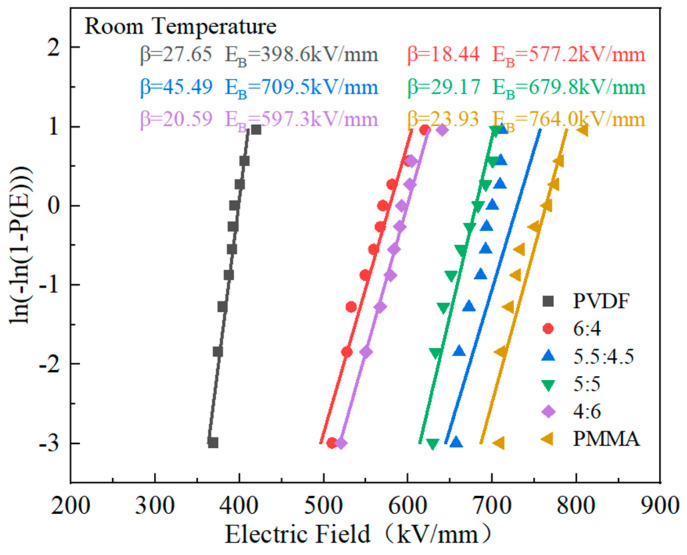
PMMA/PVDF mixes with various PMMA concentrations have varying breakdown strengths.

**Figure 9 polymers-15-02486-f009:**
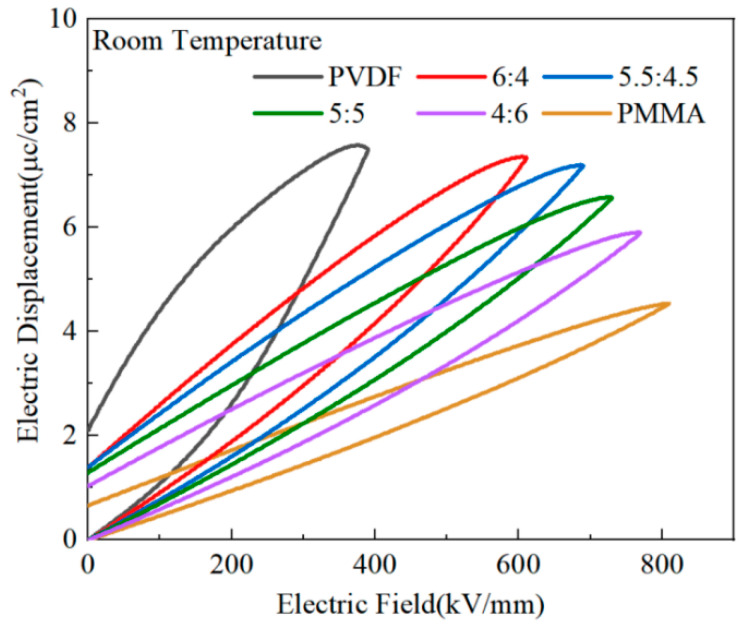
PMMA/PVDF mixes with varying amounts of PMMA were put through an electric displacement–electric field (D-E) test.

**Figure 10 polymers-15-02486-f010:**
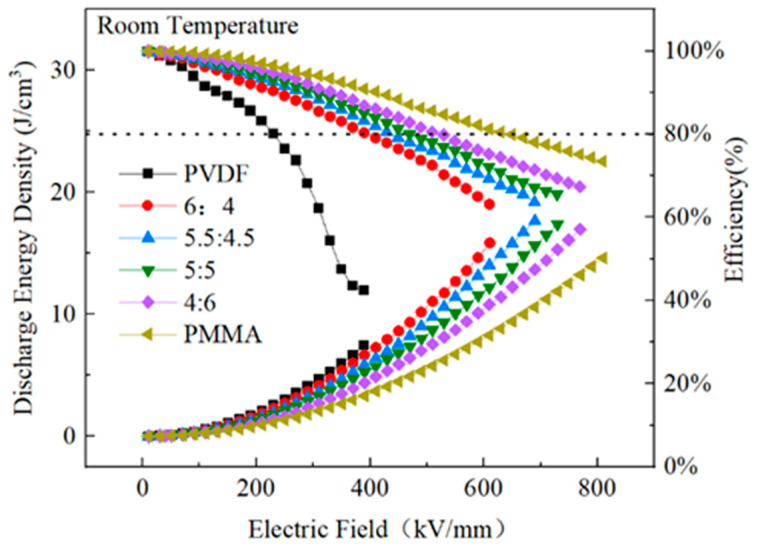
Comparing the energy storage capabilities of PMMA/PVDF mixes with varying amounts of PMMA content.

**Figure 11 polymers-15-02486-f011:**
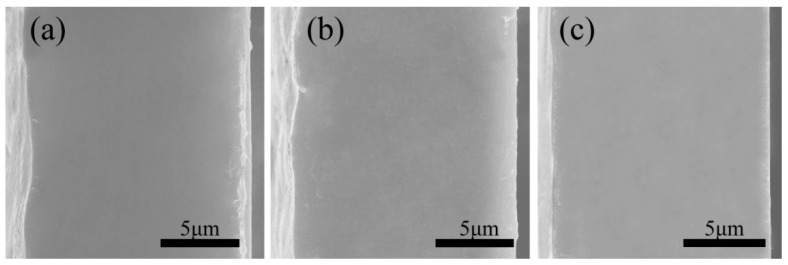
SEM photos of PVDF:PMMA = 5.5:4.5 after being subjected to three distinct heat treatments at temperatures of (**a**) 90 °C, (**b**) 120 °C, and (**c**) 150 °C, respectively.

**Figure 12 polymers-15-02486-f012:**
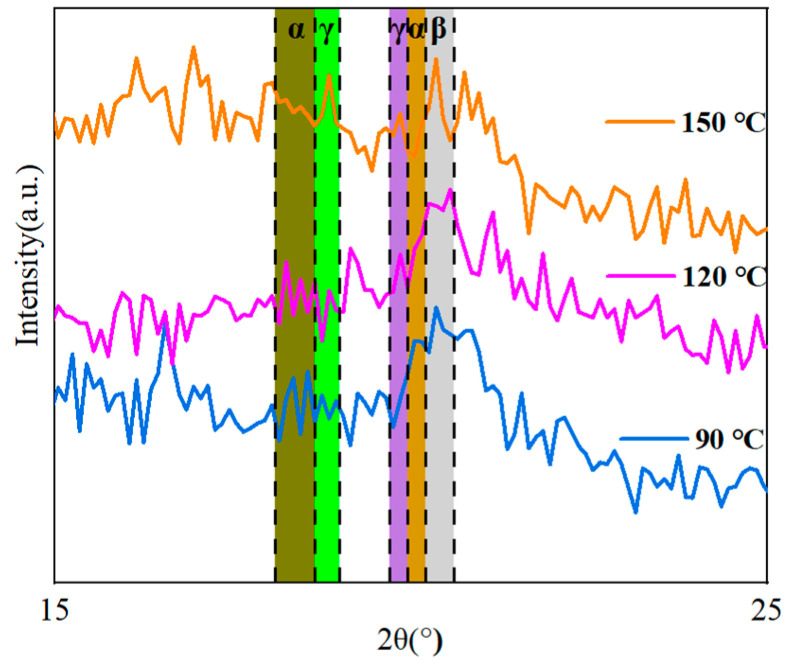
The XRD patterns of PMMA:PVDF = 5.5:4.5 blended at various temperatures during the heating process.

**Figure 13 polymers-15-02486-f013:**
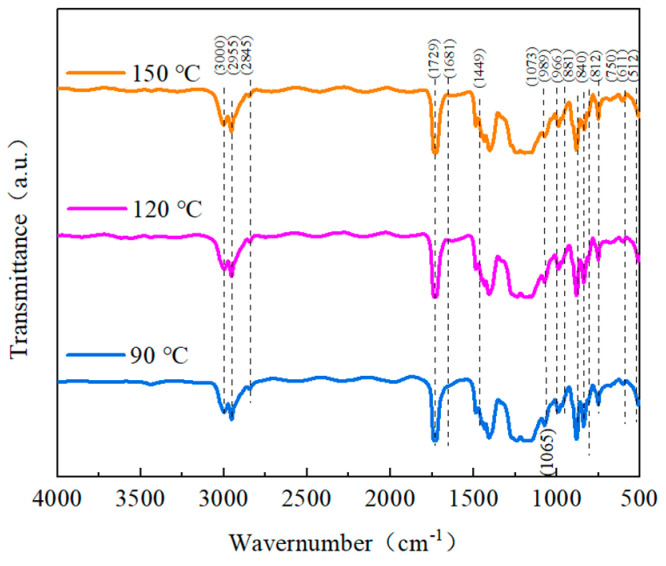
The FTIR patterns of PMMA:PVDF = 5.5:4.5 blended at different heat treatment temperatures.

**Figure 14 polymers-15-02486-f014:**
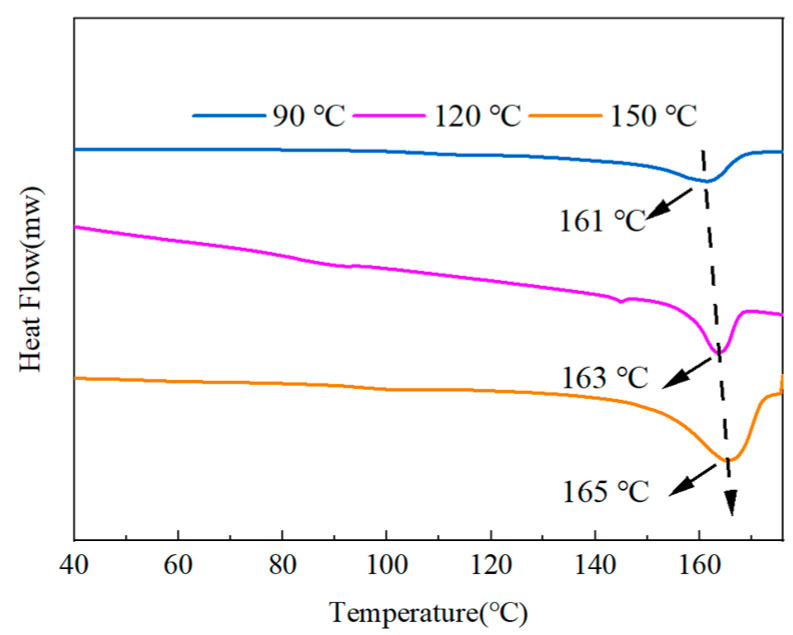
DSC curves of PMMA:PVDF = 5.5:4.5 blended at different heat treatment temperatures.

**Figure 15 polymers-15-02486-f015:**
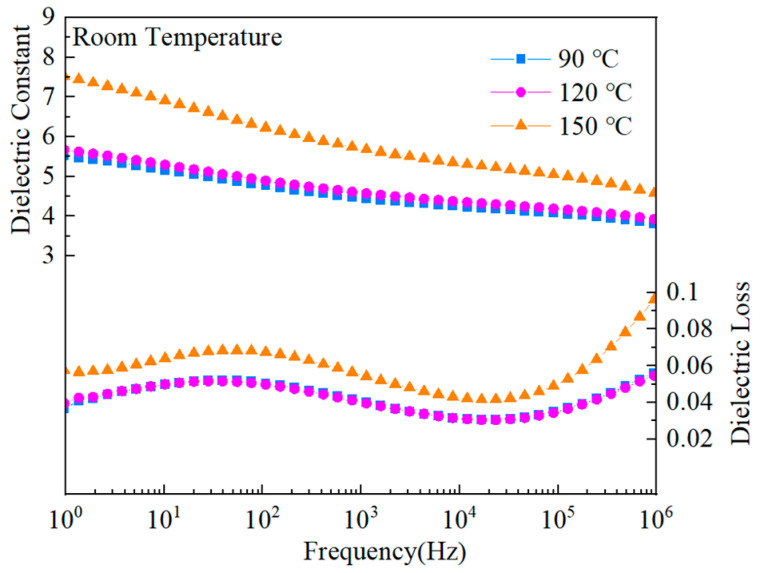
The dielectric properties of the PMMA:PVDF = 5.5:4.5 hybrid polymer at various heat treatment temperatures.

**Figure 16 polymers-15-02486-f016:**
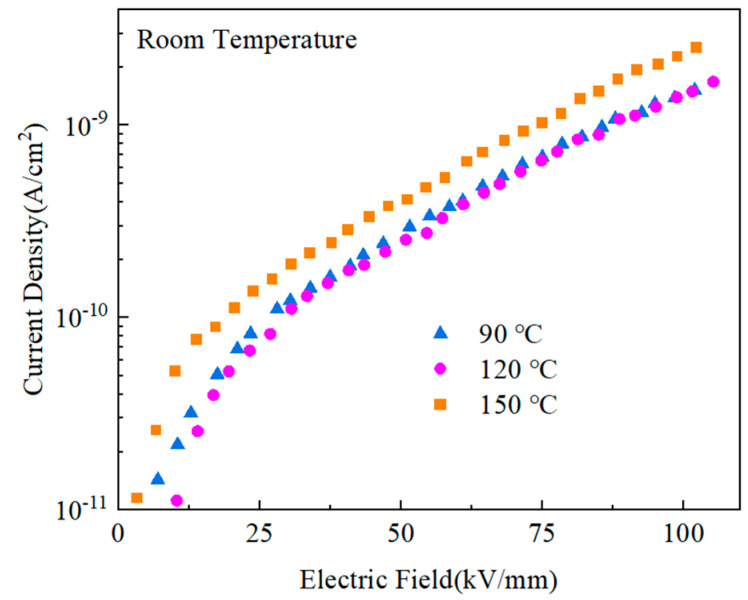
Leakage current density of PMMA:PVDF = 5.5:4.5 blended at different heat treatment temperatures.

**Figure 17 polymers-15-02486-f017:**
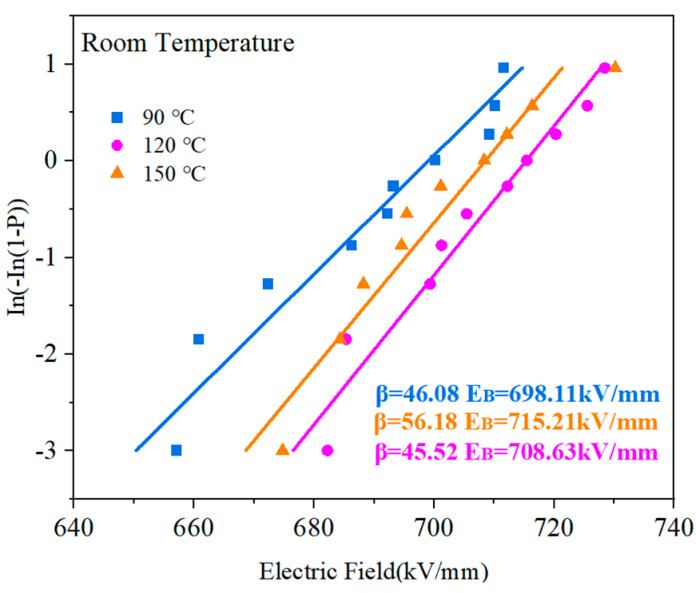
Breakdown strength of PMMA:PVDF = 5.5:4.5 blended at different heat treatment temperatures.

**Figure 18 polymers-15-02486-f018:**
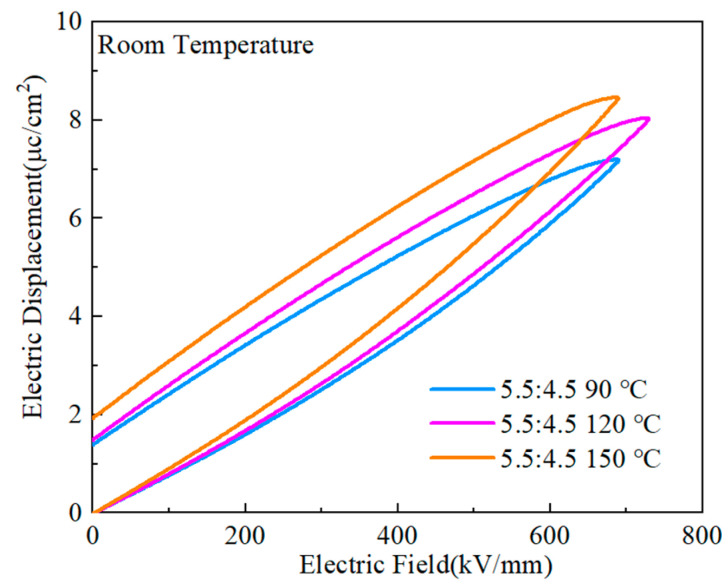
D-E curves of PMMA:PVDF = 5.5:4.5 mixed at several temperatures throughout the heat treatment process.

**Figure 19 polymers-15-02486-f019:**
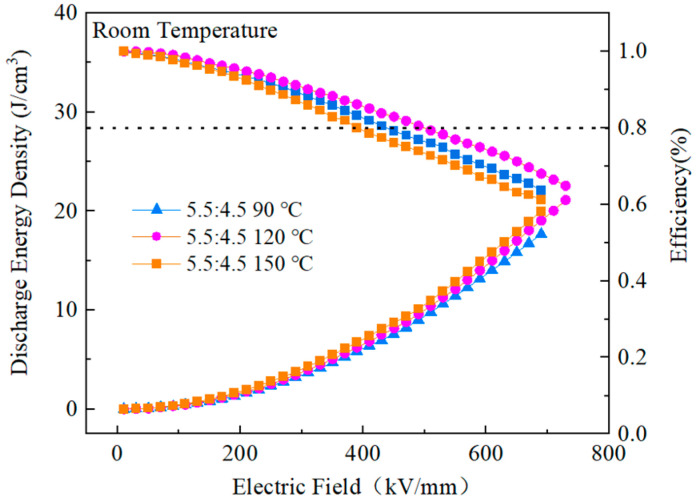
Energy storage performance of PMMA:PVDF=5.5:4.5 blended at different heat treatment temperatures.

## Data Availability

Data sharing not applicable. No new data were created or analyzed in this study. Data sharing is not applicable to this article.

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
