# Peer review of "Enhancement of Energy Storage Performance of PMMA/PVDF Composites by Changing the Crystalline Phase through Heat Treatment"

_polymers, 2023, doi:10.3390/polym15112486_

Round 1
Reviewer 1 Report
The authors describe the development and heat treatment of the dielectric polymer blend for capacitors. The manuscript can be accepted as soon as the following points will be adressed:
1) L54. Please add the units to the physical values
2) L61. Please provide the resulting formula
3) L93. The term "non-linear polymer" is unclear.
4) Section 2. Started with the SEM images. No explanation of what the samples are, how they have been obtained etc. is present in text. Please add the brief discussion concerning the sample fabrication.
5) F2 and below. Only the peak at 20.2 deg is unambiguosly seen in the XRDs. At this magnification and resolution, no additional peaks are visible. Authors should provide more resolved diffractogram or eliminate the discussion of the other peaks.
6) F3 and below. The IR vibration bans correspond to the exact vibrations of the bonds and groups of bonds. Regardless of the phase the PVDF exists in, the -CF2- vibrations are present in its IR spectrum. Vice versa, no vibrations can be attributed to the "phase" as a whole, it should be a fragment of the (macro)molecule.
7) Eq. 3 and below. "The results show..." - where are the results, Hm or Xc values? Anyway, the calculated Xc values aro non-representative since it is unable to distinguish the Hm variation caused by crystallinity of other factors, including the simple difference between Hm* of the PVDF and PMMA. This paragraph is speculative and should be removed.
8) L247. The interaction between the carbonyl (carboxyl) and methylene in not a common thing and should be proved. Moreover, the proposed hydrogen bonding between these fragments (L291) contradicts with the definition and nature of the hydrogen bonding. The authors should remove excessive and groundless speculations concerning the nature of the observed effect or provide the experimental proofs.
9) Authors should maintain the same color code for the different figures. For example, color code differs for Figs 15 (90 deg in blue) and 16 (90 deg in orange).
10) Please calculate the capacitance of your devices in F cm-3, and F g-1. The latter calculation may put the blend in positive light in comparison with the pristine PVDF since the latter one has higher density.
Authors should revise the text for typos such as missing subscript (e.g. L340), spaces (e.g. L460), repetitons (e.g. "were investigated in this investigation", L388) etc. The unit representation should be brought to a single standard, e.g. kV/mm (L349) and uC cm-2 (L350). Do not mix different representation standards in the single text.
Reviewer 2 Report
This paper investigates the performance of PVDF/PMMA blends as capacitors, and the results in figures 10&19 are interesting. However, there are many points that are not fully explained. The points listed below should be reviewed.
1) Figure 1 and Figure 11: The authors describe that the morphology is homogeneous, but why is there no crystalline structure of PVDF observed? If crystals are not visible, then even if phase separation of smaller size is occurring, it may not be observable.
2) Regarding the crystal form of PVDF, do the XRD and IR data indicate that all α, β, and γ are included? Do their proportions remain unchanged by blending? Some explanation is needed.
3) Figure 5 (DSC curve): I think that the PMMA you are using is not a crystalline polymer. Please indicate the basis for your determination that the DSC peak at 112°C is the melting point of PMMA.
4) After showing equation (3), why did you not calculate the degree of crystallinity and only discuss it qualitatively?
5) The measured temperatures for the data in Figs. 6, 7, 8, 9, 15, 16, 17, and 18 are not shown.
6) How was the cumulative failure probability shown in Figs. 8 and 17 estimated experimentally? An explanation is needed.
7) Figure 10 is probably the most important result of this study. It would be easier to understand if the authors could explain how to estimate the discharge energy density and efficiency plotted on the vertical axes from the data in Figure 9.
8) It is difficult to understand how the authors can conclude from the results in Figure 12 that "The γ-phase was steadily improved when the temperature of the heat treatment was raised from 90°C to 150°C." Please explain in more detail.
9) Regarding the IR results in Figure 13: It is difficult to see the difference in IR bands due to the difference in heat treatment temperature. It would be easier to understand the difference if the IR bands were normalized at some peaks and displayed on top of each other.
10) If the rise in loss on the high-frequency side in Figure 15 is due to the alpha relaxation (amorphous origin), why is the relaxation intensity highest in the data from the 150°C heat treatment with the highest degree of crystallinity?
Furthermore, the fact that the β relaxation appears at a lower frequency than the α relaxation is the opposite of the usual phenomenon, so it will be necessary to explain why this is so.
11) Regarding the interpretation of Figure 17: What is the basis for the statement that there is a gradual transition from the α-phase to the γ-phase as the heat treatment temperature increases? (It seems that it cannot be clearly identified by XRD or IR.) You should clearly state the data on which this interpretation is based.
Round 2
Reviewer 2 Report
The revised manuscript generally addresses the questions I asked, but there are still some points and suggestions I am not clear on, which are listed below.
1) I am not sure why SEM cannot observe the crystalline structure, but if the authors insist that it is impossible to analyze PVDF crystals by SEM, I have no particular objection.
It is well known that PVDF and PMMA are miscible in an amorphous state, so discussing SEM pictures is unimportant. I was just wondering if the amorphous part, if it is phase-separated, is visible by the authors' observation method. I hope you understand my concern.
On top of that, I cannot understand the meaning of the relevant Line 145 sentence.
“This can be deduced from the fact that the microstructure is visible in both of the cross sections.”
Do you mean that some microstructure is observed in the photograph of the cross sections? If so, showing the pictures in which the microstructure is observed would be better. You can add them in Supplementary Materials.
2) Regarding the new Figure 2: I think it would be easier to understand if the original version of the figure is also shown. In the text, there are explanations about the WAXS pattern for pure PMMA and pure PVDF.
3) Figure 5: As a minor point, it is not common to take the endothermic peak (originated from the enthalpy relaxation) as the Tg of PMMA.
Using the midpoint of the transition or the transition starting point as the Tg value will be safer.
4) Figure 17: From the explanation in the cover letter, I understand that p (lower case letter) shown on the vertical axis in Figure 17 equals P(E) (capital letter) that appeared in Eq(3), and the authors determined P(E) by statistically processing the data for n breakdown strength values from the equation P(E)=(i -0.5)/(n+0.25). The p shown on the vertical axis in Figure 17 should be the same letter as in the text. The explanation in the cover letter should also be given to general, non-specialist readers like me. It can be either in the text or in the supplementary materials.
5) Figure 10: Thank you for explaining the discharge energy density and efficiency in the cover letter. However, it would be better to explain in the text that the former is obtained from the area of the discharging curve in Figure 9 based on equation (1), and the latter is obtained from the area ratio of charging and discharging.
It is a trivial matter, but if “Efficiency” in Figure 10 is to be displayed in %, it needs to be multiplied by 100 (or "%" should be deleted.)
